# Trichostatin A-Induced Epigenetic Modifications and Their Influence on the Development of Porcine Cloned Embryos Derived from Bone Marrow–Mesenchymal Stem Cells

**DOI:** 10.3390/ijms26052359

**Published:** 2025-03-06

**Authors:** Seung-Chan Lee, Won-Jae Lee, Young-Bum Son, Yeung Bae Jin, Hyeon-Jeong Lee, Eunyeong Bok, Sangyeob Lee, Sang-Yun Lee, Chan-Hee Jo, Tae-Seok Kim, Chae-Yeon Hong, Seo-Yoon Kang, Gyu-Jin Rho, Yong-Ho Choe, Sung-Lim Lee

**Affiliations:** 1College of Veterinary Medicine, Gyeongsang National University, Jinju 52828, Republic of Korea; 2Central Research Center, Apures, Inc., Hansan-gil, Pyeongtaek-si 17792, Republic of Korea; 3College of Veterinary Medicine, Kyungpook National University, Daegu 41566, Republic of Korea; 4Department of Obstetrics, College of Veterinary Medicine, Chonnam National University, 300 Yonbongdong, Buk-gu, Gwangju 61186, Republic of Korea; 5Division of Animal Diseases & Health, National Institute of Animal Science, Rural Development Administration, Wanju-gun 55365, Republic of Korea; 6Research Institute of Life Sciences, Gyeongsang National University, Jinju 52828, Republic of Korea

**Keywords:** porcine, nuclear transfer, trichostatin A, histone acetylation, epigenetic change

## Abstract

Abnormal epigenetic reprogramming of nuclear-transferred (NT) embryos leads to the limited efficiency of producing cloned animals. Trichostatin A (TSA), a histone deacetylase inhibitor, improves NT embryo development, but its role in histone acetylation in porcine embryos cloned with mesenchymal stem cells (MSCs) is not fully understood. This study aimed to compare the effects of TSA on embryo development, histone acetylation patterns, and key epigenetic-related genes between in vitro fertilization (IVF), NT-MSC, and 40 nM TSA-treated NT-MSC (T-NT-MSC). The results demonstrated an increase in the blastocyst rate from 13.7% to 32.5% in the T-NT-MSC, and the transcription levels of *CDX2*, *NANOG*, and *IGF2R* were significantly elevated in T-NT-MSC compared to NT-MSC. TSA treatment also led to increased fluorescence intensity of acH3K9 and acH3K18 during early embryo development but did not differ in acH4K12 levels. The expression of epigenetic-related genes (*HDAC1*, *HDAC2*, *CBP*, *p300*, *DNMT3a*, and *DNMT1*) in early pre-implantation embryos followed a pattern similar to IVF embryos. In conclusion, TSA treatment improves the in vitro development of porcine embryos cloned with MSCs by increasing histone acetylation, modifying chromatin structure, and enhancing the expression of key genes, resulting in profiles similar to those of IVF embryos.

## 1. Introduction

Porcine cloning through nuclear transfer (NT) technology serves as a valuable tool for both basic and applied research, particularly in biomedicine [1]. Due to the anatomical and physiological similarities between pigs and humans, porcine cloning has been extensively studied for applications such as xenotransplantation and human genetic disease models [2]. However, the efficiency of cloning remains low, which limits its use in both agricultural and biomedical research [3]. One of the key factors contributing to the developmental failures of cloned animals is incomplete epigenetic reprogramming. This includes modifications such as histone acetylation, DNA methylation, histone phosphorylation, ADP ribosylation, ubiquitination, and sumoylation [4]. Therefore, enhancing epigenetic reprogramming may improve the developmental capacity of cloned embryos.

Among the various epigenetic mechanisms involved in nuclear reprogramming, histone modification plays a crucial role [5]. Histone acetylation and deacetylation alter chromatin structure, thereby regulating transcription [6]. Given that nuclear reprogramming occurs only for a brief period after NT, relaxing chromatin structure through histone hyperacetylation may improve cloning efficiency. Indeed, several studies have shown that abnormal epigenetic modifications, such as DNA methylation and histone modification, occur in NT embryos [7,8]. Histone acetylation at specific chromatin sites is often inversely related to DNA methylation levels, suggesting complex interdependent relationships between these two epigenetic regulatory mechanisms [9]. Therefore, successful reprogramming of histone acetylation patterns after NT is critical for converting a differentiated nucleus into a totipotent state.

Histone deacetylase inhibitors (HDACi) have been widely used to modulate histone acetylation, including compounds such as trichostatin A (TSA) [10,11,12], sodium butyrate [13], valproic acid [14], scriptaid [15], S-adenosylhomocysteine [16], m-carboxycinnamic acid bishydroxamide [17], and oxamflatin [18]. Among these, TSA is the most commonly used HDACi. Since the effects of TSA on mouse somatic cell nuclear transfer (SCNT) were first reported by Kishigami et al. [12], similar studies have been conducted in other species, including porcine [11,15]. Previous studies have highlighted the significant role of histone acetylation in epigenetic reprogramming and the development of cloned embryos [13,19]. Based on these findings, researchers have suggested the use of HDACi, such as TSA, to correct abnormal nuclear reprogramming following NT. TSA treatment has been shown to enhance the blastocyst development rate and improve blastocyst quality by inducing hyperacetylation in cloned mice [20,21]. However, the impact of TSA on the development and quality of porcine cloned embryos derived from mesenchymal stem cells (MSCs) remains unclear.

The objective of this study is to investigate whether TSA treatment can improve the development and quality of cloned porcine embryos by examining the expression patterns of genes related to embryonic development, pluripotency, imprinting, and epigenetics, and analyzing the relationship between NT embryo development and changes in histone acetylation patterns.

## 2. Results

### 2.1. Effect of TSA on the Developmental Capacity of NT Embryos Derived from Bone Marrow–MSCs (NT-BMSC)

To assess whether TSA improves the development of porcine NT embryos, a total of 672 NT embryos (TSA untreated NT-BMSC and TSA treated T-NT-BMSC) and 440 IVF embryos were analyzed for developmental rates from the 2-cell to the blastocyst stage (Figure 1). As shown in Table 1, TSA treatment did not affect the cleavage rates of NT embryos. However, the blastocyst formation rate in T-NT-BMSC embryos (32.5 ± 1.1%) was significantly (*p* < 0.05) higher than that of both IVF (14.2 ± 2.3%) and NT-BMSC (13.7 ± 1.9%) embryos (Table 1). Additionally, blastocyst cell numbers were evaluated in all three groups to further investigate the positive effects of TSA on NT embryo development. While the total cell count in T-NT-BMSC embryos was higher than in NT-BMSC embryos, the difference between the two NT groups was not statistically significant.

### 2.2. Ability of TSA to Improve Development-Related Genes in NT-BMSC Embryos

To explore the mechanism behind the enhanced development of TSA-treated cloned embryos, the expression levels of inner cell mass (ICM)/trophectoderm (TE) lineage-related genes, pluripotency-related genes, and imprinting genes at the blastocyst stage were analyzed using RT-PCR. Compared to NT-BMSC embryos, T-NT-BMSC embryos exhibited a significantly (*p* < 0.05) lower transcript level of SOX2 at the blastocyst stage. However, no differences were observed in the transcript levels of OCT4 between the NT-BMSC and T-NT-BMSC embryos. In contrast, the transcript levels of CDX2, NANOG, and IGF2R were significantly (*p* < 0.05) higher in T-NT-BMSC compared to NT-BMSC at the blastocyst stage (Figure 2).

### 2.3. Effect of TSA Treatment on Histone Acetylation Status in NT-BMSC Embryos

To investigate the effect of TSA on histone acetylation in cloned embryos, the acetylation levels of histones H3K9 (acH3K9), H3K18 (acH3K18), and H4K12 (acH4K12) were assessed in IVF, NT-BMSC, and T-NT-BMSC embryos at the 1-cell, 2-cell, 4-cell, 8-cell, and blastocyst stages via immunostaining (Figure 3, Figure 4 and Figure 5). Histone acetylation levels at different embryonic stages were quantified using corrected total cell fluorescence (CTCF) measurements. CTCF was calculated as integrated density—(area of selected cell × mean fluorescence of background readings), with Image J software (version 1.53K, NIH, Bethesda, MD, USA) used for the optical intensity analysis.

The results showed that acH3K9 levels were significantly higher in T-NT-BMSC embryos at the 1-cell, 4-cell, 8-cell, and blastocyst stages than NT-BMSC, with no significant difference observed at the 2-cell stage between the two groups. Across all groups (IVF, NT-BMSC, and T-NT-BMSC), the intensity of acH3K9 decreased rapidly from the 4-cell to the 8-cell stage, followed by an increase at the blastocyst stage (Figure 3). Similarly, acH3K18 levels were significantly higher in T-NT-BMSC embryos at the 1-cell, 2-cell, 4-cell, and blastocyst stages than in IVF and NT-BMSC embryos. There was no significant difference in acH3K18 levels at the 8-cell stage between NT-BMSC and T-NT-BMSC. As with acH3K9, the acH3K18 intensity decreased sharply from the 4-cell to the 8-cell stage, followed by an increase at the blastocyst stage (Figure 3).

In contrast, acH4K12 levels did not differ significantly between NT-BMSC and T-NT-BMSC embryos at any stage (1-cell, 2-cell, 4-cell, 8-cell, and blastocyst). The acH4K12 intensity decreased markedly at the 8-cell stage in all groups and subsequently increased at the blastocyst stage (Figure 5).

### 2.4. TSA Enhances the Expression of Genes Related to Histone Acetylation and DNA Methylation in NT-BMSC Embryos

To explore the mechanisms underlying the enhanced development of TSA-treated cloned embryos, the expression levels of genes related to histone acetylation and DNA methylation were analyzed at the 1-cell, 2-cell, 4-cell, 8-cell, and blastocyst stages using real-time RT-PCR.

As shown in Figure 6, TSA treatment significantly increased HDAC1 transcript levels in cloned embryos at the 1-cell, 4-cell, 8-cell, and blastocyst stages. Similarly, HDAC2 expression was significantly upregulated in T-NT-BMSC embryos at the 1-cell, 2-cell, 4-cell, and blastocyst stages, exhibiting a pattern similar to that observed in IVF embryos. The expression of CBP, a histone acetyltransferase-related gene, was significantly elevated in T-NT-BMSC embryos at the 1-cell, 2-cell, and blastocyst stages following TSA treatment. Additionally, p300 transcripts were significantly increased in TSA-treated cloned embryos at the 1-cell, 2-cell, 8-cell, and blastocyst stages, again showing a similar expression pattern to that of IVF embryos.

Regarding DNA methylation-related genes, DNMT3a expression was significantly upregulated in TSA-treated cloned embryos at the 1-cell, 2-cell, and blastocyst stages. DNMT1 expression was also significantly elevated at the 1-cell, 2-cell, 4-cell, and blastocyst stages, following a similar trend to that seen in IVF embryos.

## 3. Discussion

NT technique has significant applications in animal cloning, cell therapy, and conservation biology, as well as in producing cloned pigs for biomedical research due to physiological similarities with humans [22]. However, the efficiency of cloning, particularly in producing healthy pig offspring, remains suboptimal. Previous research suggests that epigenetic abnormalities, particularly histone modifications and DNA methylation, play a critical role in the limited success rate of NT cloning. BM-MSCs offer significant advantages as donor cells for NT due to their lower global DNA methylation levels and higher plasticity compared to fully differentiated somatic cells. Their epigenetic profile is more conducive to nuclear reprogramming, facilitating the activation of pluripotency genes such as OCT4, SOX2, and NANOG. Additionally, BM-MSCs exhibit increased histone acetylation marks (H3K9, H4K12, and H3K18), which are associated with transcriptionally active chromatin and improved reprogramming efficiency. Since epigenetic barriers often hinder NT embryo development, HDAC inhibitors like TSA can further enhance histone acetylation, promoting successful reprogramming and improving SCNT embryo viability [23,24,25]. Previous studies have demonstrated that BM-MSCs significantly improve blastocyst formation and offspring production in porcine SCNT compared to fibroblasts [25]. Moreover, overexpression of pluripotency-related transcription factors (TFs) such as OCT4 and SOX2 in BM-MSCs has been shown to further enhance embryo quality and cloning efficiency 26]. Undifferentiated BM-MSCs outperform both differentiated MSCs and fetal fibroblasts, making them a superior donor cell choice for NT. These findings highlight BM-MSCs’ potential to enhance cloning efficiency by providing a more favorable epigenetic environment for nuclear reprogramming [23,24,25,26]. Among these modifications, histone acetylation has garnered attention due to its role in chromatin relaxation and transcriptional activation, essential for gene expression regulation during embryogenesis [27]. High levels of histone acetylation promote a permissive chromatin state that aids in activating genes required for embryonic development.

Cloned embryos frequently exhibit atypical histone acetylation patterns, which may hinder normal development [28]. These discrepancies in histone acetylation arise due to the distinct chromatin structures between somatic and germ cells, resulting in different reprogramming requirements for cloned versus fertilized embryos. To address these challenges, HDACi such as TSA have been widely used in NT to enhance chromatin remodeling. TSA treatment has improved blastocyst formation rates across various species, including porcine and rabbit NT embryos [29,30]. TSA treatment in bovine SCNT embryos, for example, enhances pre-implantation development when used at a 50 nM concentration, illustrating the potential of HDACi to boost reprogramming efficiency [31,32]. However, its specific effects on cloned embryos derived from MSCs as donor cells, as opposed to the traditionally used fibroblasts, remain understudied.

Our study investigated the developmental impact of TSA treatment on porcine NT embryos derived from BM-MSCs. We observed that T-NT-BMSC had a significantly improved blastocyst development rate (32.5 ± 1.1%) compared to untreated embryos (13.7 ± 1.9%), underscoring TSA’s capacity to promote developmental competence in MSC-based NT embryos. However, total cell numbers did not differ significantly between treated and untreated groups, suggesting that the benefit of TSA primarily enhances blastocyst formation rather than cellular proliferation. Critical to understanding these results is the role of key pluripotency-related genes, including *OCT4*, *CDX2*, *NANOG*, and *SOX2*, which are necessary for maintaining pluripotency and supporting early embryonic development [33,34]. Contrary to reports that HDACi can elevate *OCT4* expression in blastocysts, our study observed no significant difference in *OCT4* levels between NT-BMSC and T-NT-BMSC groups. This finding aligns with previous research showing that TSA treatment does not universally increase *OCT4* expression across species or embryonic stages [30,35]. Interestingly, *OCT4* expression varies based on sex in buffalo SCNT embryos treated with TSA, where male blastocysts exhibited lower *OCT4* levels than female counterparts [32], indicating that genetic or chromosomal factors may influence *OCT4* responsiveness to TSA treatment in specific contexts. However, TSA did not significantly increase OCT4 expression in NT embryos, likely due to their reduced total cell number, particularly in the ICM, compared to IVF embryos in our study. Since ICM is the primary site of OCT4 expression, a smaller ICM may lead to lower overall OCT4-expressing cell numbers despite TSA treatment. We guess that while TSA facilitates epigenetic reprogramming, it does not directly compensate for the lower ICM cell count.

These results further demonstrated significantly higher *CDX2* expression in T-NT-BMSC blastocysts compared to untreated embryos, suggesting that TSA may influence the differentiation potential of trophoblast lineages, potentially supporting better placental formation [36]. This increase in *CDX2* parallels findings in TSA-treated bovine embryos, where elevated *CDX2* levels are associated with enhanced reprogramming outcomes [37]. Similarly, *NANOG* expression was significantly upregulated in T-NT-BMSC blastocysts, consistent with prior studies in which TSA enhanced *NANOG* expression across species, including porcine, bovine, and murine embryos [22,38]. Additionally, TSA treatment boosted *IGF2R* expression in T-NT-BMSC blastocysts, aligning these embryos more closely with IVF embryos [30]. Collectively, these findings underscore the beneficial effects of TSA on the expression of genes vital for developmental integrity.

Histone acetylation, particularly of histone H3, has a profound influence on gene regulation within cloned embryos. Histone acetylation plays a crucial role in transcriptional activation and chromatin remodeling during early embryonic development. Specifically, acH3K9 is associated with transcriptionally active chromatin and is essential for zygotic genome activation (ZGA) [39]. Similarly, acH4K12 facilitates chromatin relaxation and mediates the transition from maternal to embryonic control of development, with defects in this modification impairing nuclear reprogramming [40]. acH3K18 also contributes to chromatin accessibility and is critical for cell fate determination [41]. These histone modifications are key regulators of ZGA and nuclear reprogramming, particularly in NT embryos, where inefficient epigenetic reprogramming hinders developmental success. TSA, as a histone deacetylase inhibitor, enhances nuclear reprogramming by increasing acetylation at these critical loci, thereby improving cloning efficiency. The upregulation of acH3K9, acH4K12, and acH3K18 upon TSA treatment suggests improved chromatin accessibility and transcriptional activation, leading to enhanced developmental potential in early embryos. In our study, acH3K9 and acH3K18 levels were markedly elevated in T-NT-BMSC embryos from the 1-cell through blastocyst stages, aligning with prior findings where TSA-treated cloned embryos showed increased histone acetylation at these loci during early development [35]. Notably, although acH4K12 levels were heightened in TSA-treated embryos in other species, no significant difference was observed between NT-BMSC and T-NT-BMSC groups in our study, possibly due to differences in donor cell sources or the timing of acetylation analysis. The TSA-induced increase in histone acetylation also has implications for chromatin accessibility, facilitating the transcription of essential developmental genes. This hyperacetylation effect was evident in the enhanced acH3K9 and acH3K18 levels in our study, which may help overcome the developmental block that cloned embryos often encounter due to their abnormal epigenetic landscape. Additionally, the temporal regulation of histone acetylation appears crucial, as we observed TSA-induced histone acetylation peaking around the 4-cell stage, corresponding with ZGA.

Chromatin structure, influenced by histone acetylation, is closely linked with DNA methylation. TSA-treated porcine embryos exhibited DNA methylation patterns resembling IVF embryos, suggesting that TSA may also indirectly regulate DNA methylation through histone acetylation modulation. The interplay between DNA methylation and histone acetylation is vital for proper gene expression, with *DNMT1* maintaining DNA methylation during replication and *DNMT3a* establishing methylation de novo [3]. Our findings showed that the expression patterns of *DNMT3a* and *DNMT1* in T-NT-BMSC embryos paralleled those in IVF embryos, indicating that TSA may help re-establish DNA methylation profiles closer to physiological norms in NT embryos [42]. Taken together, our findings demonstrate that TSA treatment significantly enhances the developmental potential of NT embryos derived from BM-MSCs by promoting histone acetylation, upregulating key pluripotency-related genes, and modifying chromatin structure more closely with those of IVF embryos. These results highlight the potential of BM-MSCs as an alternative donor cell source in NT, particularly when combined with epigenetic modifiers such as TSA to improve reprogramming efficiency and developmental outcomes.

## 4. Materials and Methods

### 4.1. Chemicals and Media

All chemicals and media were purchased from the Sigma Chemical Company (St. Louis, MO, USA) and Gibco (Life Technologies, Burlington, ON, Canada), unless otherwise specified. The pH and osmolality of all media were adjusted to 7.2~7.4 and 285 ± 5 mOsm/L, respectively.

### 4.2. Preparation of Mesenchymal Stem Cells as Donor Cells

All experiments were authorized by the Animal Center for Biomedical Experimentation at Gyeongsang National University (GNU-140305). Bone marrow-derived mesenchymal stem cells (BM-MSCs) were isolated from the bone marrow extract of a 3-month-old miniature pig as previously described [5]. A total of 1 × 10^5^ cells were cultured in a 35 mm culture dish in Advanced Dulbecco’s Modified Eagle Medium (ADMEM) supplemented with 10% fetal bovine serum (FBS) and 1% penicillin–streptomycin (10,000 IU and 10,000 μg/mL, Pen-Strep) at 38 °C in a humidified atmosphere of 5% CO_2_ in the air. On reaching 100% confluent status, BM-MSCs at passage 3–5 were dissociated with 0.25% trypsin–EDTA (Invitrogen, Carlsbad, CA, USA) at 37 °C for about 3–5 min, transferred to a centrifuge tube with HEPES-buffered TCM199 supplemented with 0.3% BSA and 12 mM sorbitol, and then re-suspended for use as nuclear donor cells.

### 4.3. Oocyte Collection and In Vitro Maturation (IVM)

Porcine ovaries were obtained from the pre-pubertal gilts in a local slaughterhouse and transported to the laboratory within 2 h after their recovery. Cumulus–oocyte complexes (COCs) from follicles of 3–6 mm in diameter were aspirated using an 18-gauge needle attached to a 10 mL syringe. After being collected in Ham’s F10, selected COCs with uniform ooplasm and compact cumulus cells were then washed twice in IVM medium, which was TCM199 supplemented with 10% (*v*/*v*) porcine follicular fluid (pFF), 5% (*w*/*v*) FBS, 10 ng/mL (*w*/*v*) epidermal growth factor (EGF), 0.57 mM cysteine, 0.5 μg/mL (*w*/*v*) follicular stimulating hormone (FSH), and 0.5 μg/mL (*w*/*v*) luteinizing hormone (LH). Sets of 80 to 100 COCs were matured in 500 μL IVM medium for 22 h at 38.5 °C in a humidified atmosphere of 5% CO_2_ in the air and further matured for 22 h in IVM medium without hormone supplements.

### 4.4. Production of IVF Embryos

Matured oocytes were placed in fertilization medium (15 oocytes into each drop of 50 μL) composed of modified Tris-buffered medium (mTBM) supplemented with 0.4% BSA and 2 mM caffeine–sodium benzoate and fertilized in vitro using a previously described protocol [36] with minor modification. Fresh boar semen was provided from Gaya Artificial Insemination Center (Goseong, GN, Republic of Korea), and live sperm were prepared by the swim-up method. After adding live sperm into the mTBM droplets in a concentration of 2 × 10^4^/mL, insemination was performed for 5 h at 38.5 °C in an atmosphere of 5% CO_2_.

### 4.5. Production of NT Embryos Derived from BM-MSCs

To generate NT-BMSC (derived from BM-MSCs) and T-NT-BMSC (derived from TSA-treated BM-MSCs), cumulus cells were removed from 44 h matured oocytes by vortexing for 1 min in TL-HEPES supplemented with 0.1% (*w*/*v*) hyaluronidase. Selected metaphase-II oocytes were transferred to HEPES-buffered TCM199 supplemented with 5 μg/mL bisbenzimide (Hoechst 33342), 7.5 μg/mL cytochalasin B (CCB), 0.3% BSA, and 12 mM sorbitol covered with mineral oil. Oocytes were enucleated by aspirating the first polar body and the adjacent cytoplasm containing the metaphase plate using a beveled micropipette. Enucleation was confirmed by visualizing the karyoplast inside the pipette when viewed under ultraviolet light. A single donor cell was carefully injected into the perivitelline space of each enucleated oocyte, positioning it between the plasma membrane and the surrounding zona pellucida. The reconstructed oocytes were placed between two electrodes, overlaid with fusion medium (0.3 mM mannitol solution containing 0.1 mM MgSO_4_, 0.05 mM CaCl_2_, 0.5 mM HEPES, and 0.01% BSA), and aligned manually. Fusion and activation were induced by double pulsed with 1.2 kV/cm for 30 μ sec using a BTX Electro Square porator (ECM 830, BTX, Inc., San Diego, CA, USA). After 30 min, the reconstructed oocytes were verified for fusion using a stereomicroscope and placed into culture medium as described below.

### 4.6. In Vitro Culture of Embryos

IVF and NT embryos were cultured in porcine zygote medium 5 (PZM-5) supplemented with 3 mg/mL BSA, 20 μg/mL Eagle amino acids (EAA), 10 μg/mL nonessential amino acids (NEAA), and 0.2 mM Na-pyruvate. The embryos were incubated under mineral oil for 7 days at 38.5 °C in a humidified atmosphere of 5% CO_2_. On day 5, 10% FBS was added to the culture medium.. The rates of cleavage and blastocyst development were assessed on day 2 and day 7, respectively.

### 4.7. Treatment of TSA in NT Embryos

TSA was dissolved in dimethyl sulfoxide (DMSO). Stock solutions of the treatment group were prepared at a 1000-fold concentration and stored at −20 °C. These TSA stock solutions were diluted with culture medium (PZM-5). TSA was treated with 40 nM for 24 h after NT.

### 4.8. Embryo Collection for Assessment

Embryos in IVF, NT, and T-NT-BMSC were collected at different stages of development. Embryos at the 1-cell (30 embryos), 2-cell (20 embryos), 4-cell (15 embryos), 8-cell (10 embryos), and blastocyst stage (3 embryos) were collected at 12, 24, 48, 72, and 168 h after fusion and activation, respectively.

### 4.9. Total Cell Number in Blastocysts

To assess the quality of blastocysts, nuclear staining of the total cells of the blastocysts was performed. In brief, embryos were transferred into 1 μg/mL bisbenzimide to stain the nucleus of the cells for about 10 min, and stained whole blastocysts were mounted with the vectashield (Vector Laboratories, Inc., Burlingame, CA, USA). Samples were photographed under a fluorescence microscope.

### 4.10. Quantitative Reverse Transcription–Polymerase Chain Reaction (qRT-PCR)

The expressions of lineage differentiation-related genes (*OCT4* and *CDX2*), transcription and pluripotency-related genes (*NANOG* and *SOX2*), imprinting-related genes (*IGF2R*), and epigenetic-related genes (*HDAC1*, *HDAC2*, *CBP*, *p300*, *DNMT3a*, and *DNMT1*) were analyzed by qRT-PCR. Total RNA was purified using an RNeasy Micro Kit (Qiagen, Valencia, CA, USA) according to the manufacturer’s instructions, and residual genomic DNA was removed by using RNase Free DNase (Qiagen) treatment for 15 min. Due to the low amounts of total RNA extracted from embryos, quantification of total mRNA yield could not be measured by a UV spectrophotometer. First-strand cDNA was synthesized from the 12 μL of total RNA extracted using 8 μL of mixture with Sensicript Reverse Transcription Kit (Qiagen), 10 units of RNaseOUT Recombinant Ribonuclease Inhibitor (Invitrogen, Grand Island, NY, USA) and 1 μM oligo dT primer (Invitrogen) at 37 °C for 1 h. Quantitative RT-PCR was performed using a Rotor-Gene Q qRT-PCR machine (Qiagen) with Rotor-Gene 2X SYBR green mix (Qiagen), including 2 μL cDNA per reaction and 0.5 μM forward and reverse primers (Table 1). All experiments comprised pre-denaturation at 95 °C for 10 s, 60 °C for 6 s, and 72 °C for 4 s, a melting curve from 60 °C to 95 °C by 1 °C per sec, and cooling at 40 °C for 30 s with minor modifications in the manufacturer’s qRT-PCR program protocol. The relative quantification of gene transcription level was calculated against the reference gene, succinate dehydrogenase complex, subunit A (*SDHA*). The primers used in this study are listed in Table 2.

### 4.11. Immunofluorescence Staining of Embryos

Immunofluorescence was carried out to evaluate and compare the pattern of histone acetylation after NT. The embryos were fixed in 4% paraformaldehyde in PBS for 1 h. Subsequently, the embryos were washed in PBS and then permeabilized with 0.5% Triton X-100 in PBS for 45 min. After washing, the samples were blocked in PBS containing 3% BSA for 1 h at 38.5 °C and incubated with primary antibody at 38.5 °C (anti-acH3K9, 1:1000; anti-acH4K12, 1:1000; anti-acH3K18, 1:500). After washing in PBS containing 0.5% Tween-20, samples were incubated for 1 h with Alexa Fluor 594-conjugated goat-anti-rabbit IgG (1:200). After washing in PBS containing 0.5% Tween-20, the samples were stained with 1 ug/mL bisbenzimide for 30 min at rt. The stained samples were mounted with vectashield, and the samples were observed under a fluorescence microscope and a confocal laser scanning microscope.

The fluorescence value of global acH3K9, acH4K12, and acH3K18 in nuclei was examined using Image J software (version 1.53K, NIH, Bethesda, MD, USA). The average pixel intensity of the nuclear area was calculated by Image J software (version 1.53K, NIH, Bethesda, MD, USA), then normalized by dividing the average pixel intensity of the background areas.

### 4.12. Statistical Analysis

All experiments were repeated at least three times. Differences among the embryos were analyzed using one-way analysis of variance (ANOVA) with Tukey’s HSD post hoc test (IBM SPSS Statistics, ver. 21). Mean values and standard deviations were calculated for the outcome variables, and differences were considered significant when *p*-values were less than 0.05.

## 5. Conclusions

In conclusion, our study demonstrates that TSA treatment improves in vitro blastocyst development in cloned porcine embryos derived from MSCs by enhancing histone acetylation and modifying the expression of key epigenetic regulators. Specifically, the increased levels of acH3K9 and acH3K18 foster chromatin accessibility, enabling the activation of developmentally crucial genes and creating gene expression profiles that more closely resemble those in IVF embryos. These findings indicate that the effect of TSA on histone acetylation and epigenetic reprogramming holds promise for advancing nuclear reprogramming efficiency in cloned embryos, especially when using MSCs as donor cells, providing an enhanced framework for understanding NT-based reprogramming. The interplay between histone acetylation and DNA methylation is crucial for embryonic development. TSA has been shown to reduce DNA methylation levels compared to other inhibitors, impacting the expression of DNA methyltransferases DNMT1 and DNMT3a, which regulate methylation patterns during embryogenesis [12,43]. The current study found that DNMT3a and DNMT1 expression in T-NT-BMSC groups closely resembled that of IVF embryos at early developmental stages.

## Figures and Tables

**Figure 1 ijms-26-02359-f001:**
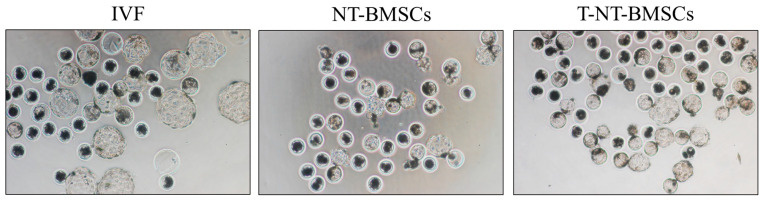
Embryos cultured in vitro for 7 days from reconstituted NT embryos, which were derived from BM-MSCs. IVF, in vitro fertilization control; NT-BMSC, NT embryos derived from BM-MSCs; T-NT-BMSC, NT embryos derived from BM-MMSCs treated with 40 nM TSA for 24 h. Magnification at ×40.

**Figure 2 ijms-26-02359-f002:**
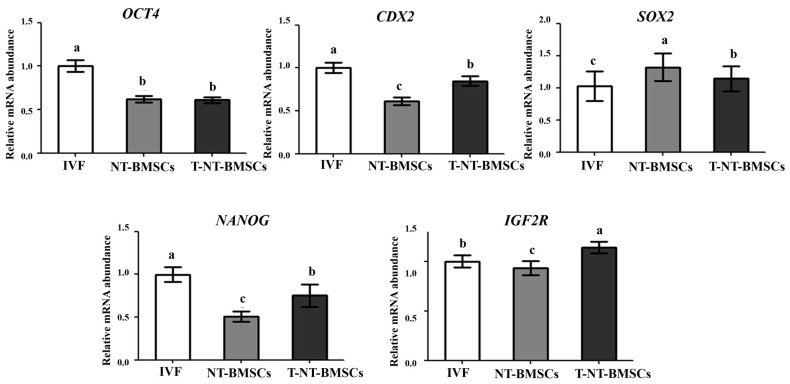
Comparative relative mRNA expression levels of transcription factors at the blastocyst stage. Porcine SDHA mRNA expression was used as the internal control, with the expression level in IVF embryos arbitrarily set to onefold. Bars labeled with different letters (a–c) indicate statistically significant differences (*p* < 0.05). Data are presented as the mean ± SD from three replicates.

**Figure 3 ijms-26-02359-f003:**
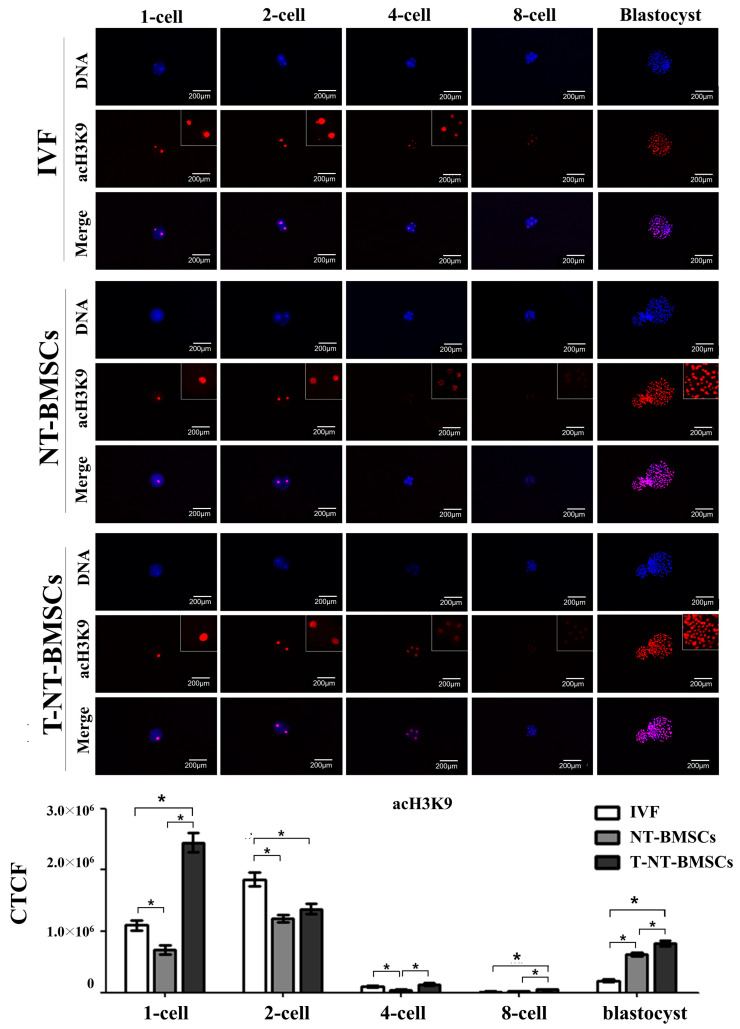
Acetylation levels of H3K9 during preimplantation development in IVF, NT-BMSCs, and T-NT-BMSCs embryos. Immunostaining of acH3K9 in embryos. Embryos were stained with anti-acH3K9 antibody (red), and DNA was counterstained with Hoechst 33342 (blue). Original magnification: 100×. Optical intensity was quantified using Image J software (version 1.53K). Data are presented as the mean ± SEM. * *p* < 0.05.

**Figure 4 ijms-26-02359-f004:**
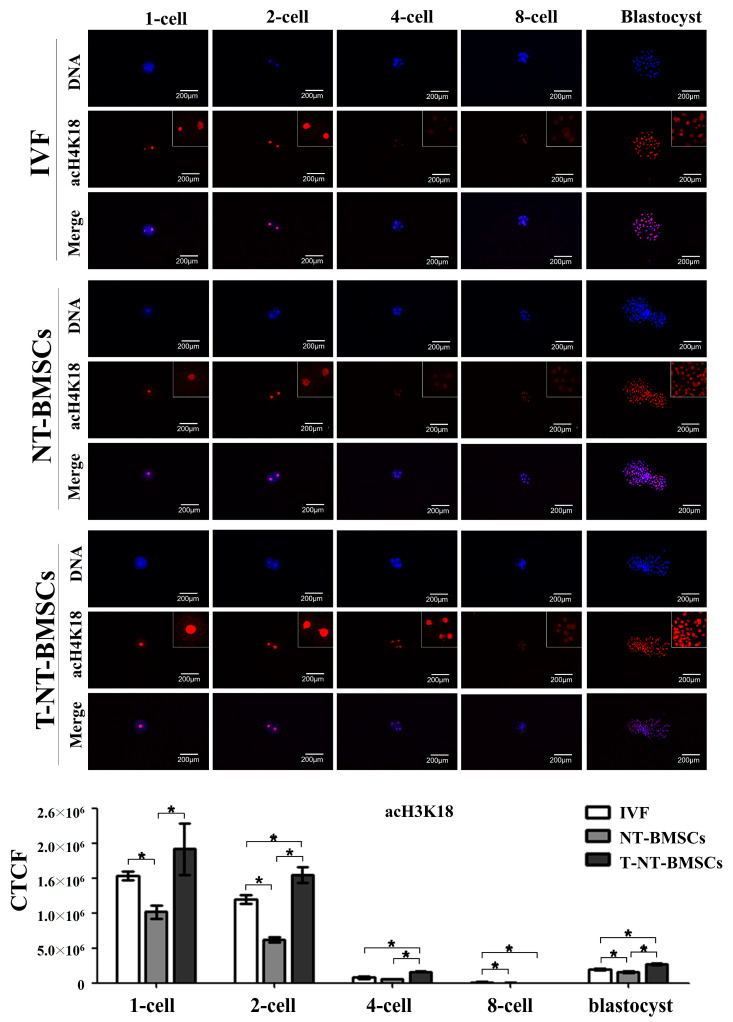
Acetylation levels of H3K18 during preimplantation development in IVF, NT-BMSC, and T-NT-BMSC embryos. Immunostaining of acH3K18 in embryos. Embryos were stained with an anti-acH3K18 antibody (red), and DNA was counterstained with Hoechst 33342 (blue). Original magnification: 100×. Optical intensity was quantified using Image J software (version 1.53K). Data are presented as the mean ± SEM. * *p* < 0.05.

**Figure 5 ijms-26-02359-f005:**
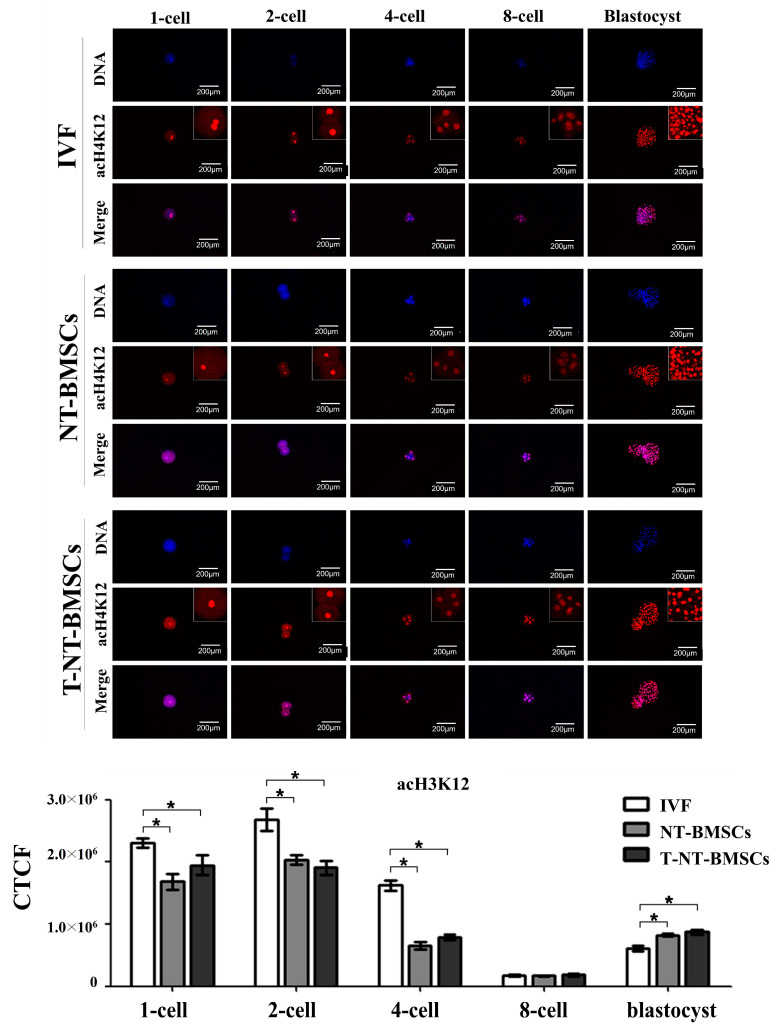
Acetylation levels of H4K12 during preimplantation development in IVF, NT-BMSCs, and T-NT-BMSCs embryos. Immunostaining of acH4K12 in embryos. Embryos were stained with an anti-acH4K12 antibody (red), and DNA was counterstained with Hoechst 33342 (blue). Original magnification: 100×. Optical intensity was quantified using Image J software (version 1.53K). Data are presented as the mean ± SEM. * *p* < 0.05.

**Figure 6 ijms-26-02359-f006:**
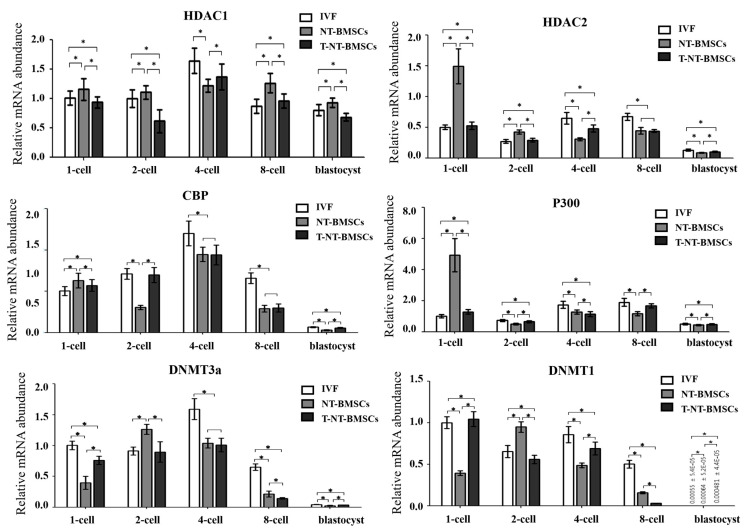
Comparative relative mRNA expression levels of *HDAC1*, *HDAC2*, *CBP*, *p300*, *DNMT3a*, and *DNMT1* at different embryo stages. Porcine SDHA mRNA expression was used as the internal control, with the expression level in IVF embryos arbitrarily set to onefold. Data are presented as the mean ± SD from three replicates. * *p* < 0.05.

**Table 1 ijms-26-02359-t001:** Effect of TSA on the development of porcine embryos cloned with BM-MSCs.

Group	No. of Reconstructed Embryos	Day 2 No. of Cleaved Embryos (Mean% ± SEM)	Day 7 Blastocysts	
			No. of Blastocysts (Mean% ± SEM)	No. of Total Cells (Mean ± SEM)
^1^ IVF	440	267 (59.2 ± 10.5) ^a^	66 (14.2 ± 2.3) ^a^	61.5 ± 5.9
^2^ NT-BMSC	347	249 (71.3 ± 4.6) ^b^	48 (13.7 ± 1.9) ^a^	61.6 ± 4.7
^3^ T-NT-BMSC	325	227 (69.0 ± 2.4) ^b^	105 (32.5 ± 1.1) ^b^	66.4 ± 6.8

Statistically analyzed data of embryos on days 2 and 7 after in vitro culture are represented as mean% ± SEM. SEM, standard error of the mean; ^1^ IVF, in vitro fertilization control (three replicates); ^2^ NT, embryos cloned with BM-MSCs (six replicates); ^3^ NT-TSA, embryos cloned with BM-MSCs treated with 40 nM TSA for 24 h (five replicates). ^a,b^ *p* < 0.05.

**Table 2 ijms-26-02359-t002:** Details of primers used for qRT-PCR.

Gene	Primer Sequence (5′-3′)	GenBank Accession Number
*OCT4*	F: AGT CCC AGG ACA TCA AAG CG R: CCT TCC CAA AGA GAA CCC CC	NM_001113060.1
*SOX2*	F: CGC AGA CCT ACA TGA ACG R: TCG GAC TTG ACC ACT GAG	NM_001123197.1
*NANOG*	F: CGA AGC ATC CAT TTC CAG CG R: TCG AGG GTC TCA GCA GAT GA	NM_001129971.1
*CDX2*	F: TCT TCC CTG CAA GGC TCG GT R: GAC GGT GGG GTT TAG CAC GC	NM_001278769.1
*IGF2R*	F: CGC TCT CTG CCT CTA GCA GT R: CCT ACA CCC CAA GTC TGG AA	XM_013992438.1
*HDAC1*	F: CCT GGA CAC GGA GAT CCC TA R: GGG CAG CAT TCT CAG GTT CT	XM_013999116.1
*HDAC2*	F: TGC AGA GAT TTA ATG TTG GA R: GTT GTC GGT TTA ACT TCA CA	XM_001925318.5
*DNMT1*	F: AGA ATT ATC AGA GGA GGG CTA CC R: CAT TCA CTT CCC GAC TGA AAG C	NM_001032355
*DNMT3A*	F: CTG AGA AGC CCA AG TCA AG R: CAG CAG ATG GTG CAG TAG GA	NM_001123197.1
*CBP*	F: GTC TGG CTC ATG TTC AAC AAC G R: GTA CTT TCG TCC ACA GCA ATA CC	XM_003354647
*P300*	F: TTT CTT CCT CAG GCT CAG TTC C R: AGG CAT TAT AGG AGA GTT CAC AGG	XM_001929213
*SDHA*	F: CAC ACG CTT TCC TAT GTC GAT G R: TGG CAC AGT CAG CTT CAT TC	XM_003362140.1

## Data Availability

Data can be made available upon reasonable request from the corresponding authors.

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
