# Peer review of "Trichostatin A-Induced Epigenetic Modifications and Their Influence on the Development of Porcine Cloned Embryos Derived from Bone Marrow–Mesenchymal Stem Cells"

_ijms, 2025, doi:10.3390/ijms26052359_

Round 1

Reviewer 1 Report

Comments and Suggestions for Authors

Conducted research evaluated trichostatin A (TSA) influence as epigenetic modifier on the development of porcine cloned embryos with the help of bone marrow-mesenchymal stem cells (BM-MSCs). I found the presented idea interesting, especially in the sense of TSA potential for epigenetic improvement of obtained embryos. The interesting point is the application of BM-MSCs in the mentioned approach, which potentially enhances the development of nuclear transferred (NT)-embryos. The study comprised the development of porcine embryos using in vitro fertilization (IVF), NT-MSC, and TSA-treated NT-MSC embryos group (T-NT-MSC) to show the TSA impact on the success rate of developed porcine embryos and the underlying molecular mechanisms of that epigenetic improvement. The molecular evaluation comprised the assessment of relative mRNA abundance in loci of genes encoding histone deacetylases, and such relevant to embryonic development, pluripotency, imprinting and the maintenance of inherited and de-novo methylation patterns.  The significant part of that research included also fluorescence immunostaining results of particular histone modifications. Conducted research showed the high impact of TSA on the development of cloned porcine embryos.

Despite its high value, the manuscript requires certain points to be addressed.

Rows 75 to 78 – please describe in more detail the rationale of conducted research in the context of “the development and quality of porcine cloned embryos derived from mesenchymal stem cells”. Why is this approach interesting? What are the advantages?

Row 83 - The “NT-BMSC” embryos  - expand the BMSC abbreviation

Row 105 – “ICM/TE lineage-related genes” - expand the ICM/TE abbreviation

Row 256 - Discussed results need a small summary to provide the global picture of conducted research. (e.g. What is the relevance of observed epigenetic alterations in the context of BM-MSCs application in the NT-embryo development.)

Row 295 -  Production of NT embryos derived from BM-MSCs What is the function of BM-MSCs in this method? Do they provide support for developing embryos?

Row 304 – “A single donor cell was placed into the perivitelline space of each enucleated oocyte.” Please clarify the sentence.

Row 367 – please describe the potential role of acH3K9, acH4K12, and acH3K18 modifications in embryo epigenetic reprogramming. Why they were chosen for the evaluation of TSA reprogramming effect.

Reviewer 2 Report

Comments and Suggestions for Authors

The article "Trichostatin A-induced epigenetic modifications and their influence on the development of porcine cloned embryos derived from bone marrow-mesenchymal stem cells" addresses a significant topic concerning the effects of trichostatin A (TSA) on the epigenetic reprogramming of cloned porcine embryos.

The study is well-designed and makes an important contribution to understanding the mechanisms regulating the development of nuclear transfer (NT) embryos. The results are clearly presented, and the methods used, such as immunofluorescence and RT-qPCR, allow for a reliable analysis of epigenetic modifications.

The authors provide strong evidence that TSA enhances the development of NT embryos derived from bone marrow mesenchymal stem cells (BMSC) by increasing histone acetylation levels, leading to improved chromatin reprogramming. A significant increase in the blastocyst formation rate was observed in the TSA-treated group compared to untreated NT embryos, which is consistent with previous reports on the effects of histone deacetylase inhibitors on embryo development in other species. Additionally, the expression of key genes associated with pluripotency and epigenetics, such as CDX2, NANOG, and IGF2R, was significantly elevated following TSA treatment, suggesting that this compound supports nuclear reprogramming to a state more similar to that of IVF embryos.

From a methodological perspective, the study was conducted rigorously, and the description of the techniques used is sufficiently detailed to allow replication. However, it is recommended to clarify the number of biological replicates in each experiment, particularly concerning gene expression analyses and immunofluorescence studies. Furthermore, it would be beneficial to include a control group in which NT embryos are treated with DMSO alone (the solvent for TSA) to rule out potential effects of this factor on the results.

One of the intriguing findings of this study is the lack of a significant effect of TSA on H4K12 acetylation levels, which differs from previous reports in other animal models. The authors could discuss this aspect more extensively in the context of species-specific epigenetic mechanisms in pigs compared to other organisms. Additionally, the discussion could be expanded to explore potential mechanisms underlying the observed changes in NANOG and CDX2 expression, particularly regarding their role in embryonic development and possible interactions with TSA.

Regarding language and style, the manuscript is well-written, though some minor editorial corrections would improve clarity. More precise wording is recommended in certain conclusions, such as those concerning the effect of TSA on OCT4 expression, which, unlike other genes, did not show significant changes. The authors could emphasize that these results align with some previous studies while also discussing possible differences arising from the animal model or experimental conditions.

In summary, this manuscript represents a valuable contribution to research on epigenetic reprogramming in cloned embryos and can be accepted for publication after minor revisions. The most important recommendations include refining the methodology section, expanding the discussion to consider alternative mechanisms of TSA action, and making minor linguistic corrections.

Comments on the Quality of English Language

Regarding language and style, the manuscript is well-written, though some minor editorial corrections would improve clarity. More precise wording is recommended in certain conclusions, such as those concerning the effect of TSA on OCT4 expression, which, unlike other genes, did not show significant changes. 
